# Scaling Riemannian Diffusion Models

**Aaron Lou, Minkai Xu, Adam Farris, Stefano Ermon**
Stanford University
`{aaronlou, minkai, adfarris, ermon}@stanford.edu`

## Abstract

Riemannian diffusion models draw inspiration from standard Euclidean space diffusion models to learn distributions on general manifolds. Unfortunately, the additional geometric complexity renders the diffusion transition term inexpressible in closed form, so prior methods resort to imprecise approximations of the score matching training objective that degrade performance and preclude applications in high dimensions. In this work, we reexamine these approximations and propose several practical improvements. Our key observation is that most relevant manifolds are symmetric spaces, which are much more amenable to computation. By leveraging and combining various ansätze, we can quickly compute relevant quantities to high precision. On low dimensional datasets, our correction produces a noticeable improvement, allowing diffusion to compete with other methods. Additionally, we show that our method enables us to scale to high dimensional tasks on nontrivial manifolds. In particular, we model QCD densities on $SU(n)$ lattices and contrastively learned embeddings on high dimensional hyperspheres.

## 1 Introduction

By learning to faithfully capture high-dimensional probability distributions, modern deep generative models have transformed countless fields such as computer vision [19] and natural language processing [41]. However, these models are built primarily for geometrically simple data spaces, such as Euclidean space for images and discrete space for text. For many applications such as protein structure prediction [13], contrastive learning [12], and high energy physics [6], the support of the data distribution is instead a Riemannian manifold such as the sphere or torus. Here, naïvely applying a standard generative model on the ambient space results in poor performance as it doesn't properly incorporate the geometric inductive bias and can suffer from singularities [7].

As such, a longstanding goal within Geometric Deep Learning has been the development of principled, general, and scalable generative models on manifolds [8, 18]. One promising method is the Riemannian Diffusion Model [4, 25], the natural generalization of standard Euclidean space score-based diffusion models [46, 48]. These learn to reverse a diffusion process on a manifold–in particular, the heat equation–through Riemannian score matching methods. While this approach is principled and general, it is not scalable. In particular, the additional geometric complexity renders the denoising score matching loss intractable. Because of this, previous work resorts to inaccurate approximations or sliced score matching [47], but these degrade performance and can't be scaled to high dimensions. We emphasize that this fundamental problem causes Riemannian Diffusion Models to fail for even trivial distributions on high dimensional manifolds, which limits their applicability to relatively simple low-dimensional examples.

In our work, we propose several improvements to Riemannian Diffusion Models to stabilize their performance and enable scaling to high dimensions. In particular, we reexamine the heat kernel [21], which is the core building block for the denoising score matching objective. To enable denoising

---

Code found at https://github.com/louaaron/Scaling-Riemannian-Diffusion

37th Conference on Neural Information Processing Systems (NeurIPS 2023).

score matching, one needs to be able to sample from and compute the gradient of the logarithm of the heat kernel efficiently. This can be done trivially in Euclidean space as the heat kernel is a Gaussian distribution, but doing this is effectively intractable for general manifolds. By restricting our analysis to Riemannian symmetric spaces [9, 29], which are a class of a manifold with a special structure, we can make substantial improvements. We leverage this additional structure to quickly and precisely compute heat kernel quantities, allowing us to scale up Riemannian Diffusion Models to high dimensional real-world tasks. Furthermore, since almost all manifolds that practitioners work with are (or are diffeomorphic to) Riemannian symmetric spaces, our improvements are generalizable and not task-specific. Concretely our contributions are:

- **We present a generalized strategy for numerically computing the heat kernel on Riemannian symmetric spaces in the context of denoising score matching.** In particular, we adapt known heat kernel techniques to our more specific problem, allowing us to quickly, accurately, and stably train with denoising score matching.

- **We show how to exactly sample from the heat kernel using our quick computations above.** In particular, we show how our exact heat kernel computation enables fast simulation free techniques on simple manifolds. Furthermore, we develop a sampling method based on the probability flow ODE and show that this can be quickly computed with standard ODE solvers on the maximal torus of the manifold.

- **We empirically demonstrate that our improved Riemannian Diffusion Models improve performance and scale to high dimensional real world tasks.** For example, we can faithfully learn Wilson action on $4 \times 4$ $SU(3)$ lattices (128 dimensions). Furthermore, when applied to contrastively learned hyperspherical embeddings (127 dimensions), our method enables better model interpretability by recovering the collapsed projection head representations. To the best of our knowledge, this is the first example where differential equation-based manifold generative models have scaled to real world tasks with hundreds of dimensions.

## 2 Background

### 2.1 Diffusion Models

Diffusion models on $\mathbb{R}^d$ are defined through stochastic differential equations [24, 45, 48]. Given an initial data distribution $p_0$ on $\mathbb{R}^d$, samples $\mathbf{x}_0 \sim \mathbb{R}^d$ are perturbed with a stochastic differential equation[33]

$$d\mathbf{x}_t = \mathbf{f}(\mathbf{x}_t, t)dt + g(t)d\mathbf{B}_t \qquad (1)$$

where $\mathbf{f}$ and $g$ are fixed drift and diffusion coefficients, respectively. The time varying distributions $p_t$ (defined by $\mathbf{x}_t$) evolves according to the Fokker-Planck Equation

$$\frac{\partial}{\partial t}p_t(\mathbf{x}) = -\operatorname{div}(p_t(\mathbf{x})\mathbf{f}(\mathbf{x}, t)) + \frac{g(t)^2}{2}\Delta_x p_t(\mathbf{x}) \qquad (2)$$

and approaches a limiting distribution $\pi \approx p_T$, which is normally a simple distribution like a Gaussian $\mathcal{N}(0, \sigma_T^2 I)$ through carefully chosen $\mathbf{f}$ and $g$. Our SDE has a corresponding reversed SDE

$$d\mathbf{x}_t = (\mathbf{f}(\mathbf{x}_t, t) - g(t)^2 \nabla_x \log p_t(\mathbf{x}_t))dt + g(t)d\overline{\mathbf{B}}_t \qquad (3)$$

which maps $p_T$ back to $p_0$. Motivated by this correspondence, diffusion models approximate the score function $\nabla_x \log p_t(\mathbf{x})$ using a neural network $\mathbf{s}_\theta(\mathbf{x}, t)$. To do this, one minimizes the score matching loss [27], which is weighted by constants $\lambda_t$:

$$\mathbb{E}_{t, \mathbf{x}_t \sim p_t} \lambda_t \left\| \mathbf{s}_\theta(\mathbf{x}_t, t) - \nabla_x \log p_t(\mathbf{x}_t) \right\|^2 \qquad (4)$$

Since this loss is intractable due to the unknown $\nabla_x \log p_t(\mathbf{x}_t)$, we instead use an alternative form of the loss. One such loss is the implicit score matching loss[27]:

$$\mathbb{E}_{t, \mathbf{x}_t \sim p_t} \lambda_t \left[ \operatorname{div}(\mathbf{s}_\theta)(\mathbf{x}_t, t) + \frac{1}{2} \left\| \mathbf{s}_\theta(\mathbf{x}_t, t) \right\|^2 \right] \qquad (5)$$

which normally is estimated using sliced score matching/Hutchinson's trace estimator[26, 47]:

$$\mathbb{E}_{t, \epsilon, \mathbf{x}_t \sim p_t} \lambda_t \left[ \epsilon^\top D_x \mathbf{s}_\theta(\mathbf{x}_t, t)\epsilon + \frac{1}{2} \left\| \mathbf{s}_\theta(\mathbf{x}_t, t) \right\|^2 \right] \qquad (6)$$

where $\epsilon$ is drawn over some $0$ mean and identity covariance distribution like the standard normal distribution or the Rademacher distribution. Unfortunately, the added variance from the $\epsilon$ normally renders this loss unworkable in high dimensions[46], so practitioners instead use the denoising score matching loss[51]

$$\mathbb{E}_{t,\mathbf{x}_0 \sim p_0, \mathbf{x}_t \sim p_t(\cdot | \mathbf{x}_0)} \lambda_t \left\| \mathbf{s}_\theta(\mathbf{x}_t, t) - \nabla_x \log p_t(\mathbf{x}_t | \mathbf{x}_0) \right\|^2 \tag{7}$$

where $p_t(\mathbf{x}_t|\mathbf{x}_0)$ is derived from the SDE in Equation 1 and is normally tractable. Once $\mathbf{s}_\theta(\mathbf{x}_t, t)$ is learned, we can construct a generative model by first sampling $\mathbf{x}_T \sim \pi \approx p_T$ and solving the generative SDE from $t = T$ to $t = 0$:

$$\mathrm{d}\mathbf{x}_t = (\mathbf{f}(\mathbf{x}_t, t) - g^2(t)\mathbf{s}_\theta(\mathbf{x}_t, t))\mathrm{d}t + g(t)\mathrm{d}\overline{\mathbf{B}}_t \tag{8}$$

Furthermore, there exists a corresponding "probability flow ODE" [48]

$$\mathrm{d}\mathbf{x}_t = (\mathbf{f}(\mathbf{x}_t, t) - \frac{g(t)^2}{2}\nabla_x \log p_t(\mathbf{x}_t))\mathrm{d}t \tag{9}$$

that has the same evolution of $p_t$ as the SDE in Equation 1. This can be approximated using our score network $\mathbf{s}_\theta$ to get a Neural ODE [11]

$$\mathrm{d}\mathbf{x}_t = (\mathbf{f}(\mathbf{x}_t, t) - \frac{g(t)^2}{2}\mathbf{s}_\theta(\mathbf{x}_t, t))\mathrm{d}t \tag{10}$$

which can be used to evaluate exact likelihoods of the data [20].

## 2.2 Riemannian Diffusion Models

To generalize diffusion models to $d$-dimensional Riemannian manifolds $\mathcal{M}$, which we assume to be compact, connected, and isometrically embedded in Euclidean space, one adapts the existing machinery to the geometrically more complex space [4]. Riemannian manifolds are deeply analytic constructs, so Euclidean space operations like vector fields $\mathbf{v}$, gradients $\nabla$, and Brownian motion $\mathbf{B}_t$ have natural analogues for $\mathcal{M}$. This allows one to mostly port over the diffusion model machinery from Euclidean space. Here, we highlight some of the core differences.

**The forward SDE is the heat equation.** The particle dynamics typically follow a Brownian motion:

$$\mathrm{d}\mathbf{x}_t = g(t)\mathrm{d}\mathbf{B}_t \tag{11}$$

Unlike in the Euclidean case, $p_t$ approaches the uniform distribution $\mathcal{U}_\mathcal{M}$ as $t \to \infty$ (in practice, this convergence is fast, getting within numerical precision for $t \approx 5$).

**The transition density has no closed form.** Despite the fact that we work with the most simple SDE, the transition kernel defined by the manifold heat equation $p_t(x_t|x_0)$ has no closed form. This transition kernel is known as the heat kernel, which satisfies Equation 2 with the additional condition that, as $t \to 0$, the kernel approaches $\delta_{x_0}$. We will denote this by $K_\mathcal{M}(x_t|x_0, t)$, and we highlight that, when $\mathcal{M}$ is $\mathbb{R}^d$, this corresponds to a Gaussian and is easy to work with.

This has several major consequences which cause prior work to favor sliced score matching over denoising score matching. First, to sample a point $x \sim K_\mathcal{M}(\cdot|x_0, t)$, one must simulate a Geodesic Random Walk with the Riemannian exponential map $\exp$:

$$x_{t+\Delta t} = \exp_x(\sqrt{\Delta t}z) \quad z \sim \mathcal{N}(0, I_d) \in T_x\mathcal{M} \tag{12}$$

Additionally, to calculate $K_\mathcal{M}(\cdot|x_0, t)$ or $\nabla \log K_\mathcal{M}(x|x_0, t)$, one must use eigenfuncions $f_i$ which satisfy $\Delta f_i = -\lambda_i f_i$. These $f_i$ form an orthonormal basis for all $L^2$ function on $\mathcal{M}$, allowing us to write the heat kernel with an infinite sum:

$$K_\mathcal{M}^{\mathrm{EF}}(x|x_0, t) = \sum_{i=0}^{\infty} e^{-\lambda_i t} f_i(x_0) f_i(x) \tag{13}$$

Additionally, previous work has also explored the use of the Varadhan approximation for small values of $t$ (which uses the Riemannian logarithmic map $\log$) [43]:

$$K_\mathcal{M}(x|x_0, t) \approx \mathcal{N}(0, \sqrt{2t})(\mathrm{dist}_\mathcal{M}(x_0, x)) \implies \nabla_x \log K_\mathcal{M}(x|x_0, t) \approx \frac{1}{2t}\log_x(x_0) \tag{14}$$

# 3 Method

The key problem with applying denoising score matching in practice is that the heat kernel computation is too expensive and inaccurate. Notably, simulating a geodesic random walk is expensive since it requires many exponential maps. Furthermore, the eigenfunction expansion in Equation 13 requires more and more eigenfunctions (numbering in the tens of thousands) as $t \to 0$. Worse still, these eigenfunctions formulas are not well-known for most manifolds, and, even when explicit formulas exist, they can be numerically unstable (like in the case of $S^n$). One possible way to alleviate this is to use Varadhan's approximation for small $t$, but this is also unreliable except for very small $t$.

To remedy this issue, we instead consider the case of Riemannian symmetric spaces [22]. We emphasize that most manifold generative modeling applications already model on Riemannian Symmetric Spaces like the sphere, torus, or Lie Groups, so we do not lose applicability by restricting our attention here. Furthermore, for surfaces (which have appeared as generative modeling test tasks in the literature [42]), one can always define a generative model by mapping the data points to $S^2$, learning a generative model there, and mapping back [14]. We empathize that, outside of these two examples, we are unaware of any other manifolds which have been used for Riemannian generative modeling tasks. For this very reason, we will also focus primarily on the compact case, since the irreducible (base case) noncompact symmetric spaces are all diffeomorphic to Euclidean space, although we discuss the extensions of our method in Appendix A.2.

## 3.1 Heat Kernels on Riemannian Symmetric Spaces

In this section, we will define Riemannian Symmetric Spaces and showcase their relationship with the heat kernel. We empathize that our exposition is neither rigorous nor fully defines all terms, although it provides a general intuition with which to build our calculations. We urge interested readers to consult a book [22] or monograph [9] for a full treatment of the subject.

**Definition 3.1.** *A Riemannian Symmetric Space is a Riemannian manifold such that, for all points $x \in \mathcal{M}$, there exists a local isometry $s$ s.t. $s(x) = x$ and $D_x s = -\mathrm{id}_{T_x \mathcal{M}}$.*

Intuitively, this condition means that, at each point, the manifold "looks the same" in all directions, which simplifies the analysis. While this is defined as a local condition, it also has a global consequence: all Riemannian symmetric spaces are quotients $G/K$ for Lie Groups $G$ and compact isotropic subgroups $K$. Importantly, most Riemannian manifolds that are used in practice are symmetric:

**Examples.** *Lie Groups $G \cong G/\{e\}$ (where $\{e\}$ is the trivial Lie Group), the sphere $S^n \cong SO(n+1)/SO(n)$, and hyperbolic space $H^n \cong SO(n,1)/O(n)$ are all symmetric spaces.*

On symmetric spaces, one can define a special structure called the maximal torus:

**Definition 3.2** (Maximal Torus). *A maximal torus $T$ is a compact, connected, and abelian subgroup of $G$ that is not contained in any other such subgroup. For symmetric spaces $G/K$, the maximal torus is defined by quotienting out the maximal torus of $G$. Importantly, the maximal torus is isomorphic to a standard torus $T^n \approx (S^1)^n$.*

Intuitively, the maximal torus forms the "most stable" part of the symmetric space, and it shows up as a natural object when performing analysis due to its isometry with the standard flat torus.

**Examples.** *For the sphere $S^n$, a maximal torus is any great circle. For the Lie group of unitary matrix $U(n)$, the maximal torus is the set of all diagonal matrices $\{\mathrm{diag}(e^{i\theta_1}, \ldots, e^{i\theta_n}) : \theta_k \in [0, 2\pi)\}$.*

To work with the maximal torus algebraically (as we shall do in our definition of the heat kernel), we instead analyze the root system.

**Definition 3.3** (Roots of a Maximal Torus). *We define a root $\alpha$ be a vector (really a covector) that corresponds with a character $\chi$ of $T$ s.t.*

$$\chi(\exp X) = \exp(2\pi i(\alpha \cdot X)) \tag{15}$$

*We define the set $R^+$ to be the set of all positive roots. The multiplicity $m_\alpha$ of a root $\alpha$ is the dimension of the eigenspace over the Lie algebra $\mathfrak{G}$*

$$\{Z \in \mathfrak{G} : [H, Z] = \pi i \lambda(H) Z \ \forall H\} \tag{16}$$

**Examples.** *For all spheres dimension $n$, there is only one root of multiplicity $n-1$, which corresponds (roughly) to measuring the angle on the maximal torus/great circle.*

Notably, the heat kernel is invariant on the maximal torus. This allows us to write out the formula for the heat kernel on the general manifold as a simplified formula on the maximal torus

**Proposition 3.4** (Heat Kernel Reduces on Maximal Torus). *The Laplace-Beltrami operator on $\mathcal{M}$ (the manifold generalization of the standard Laplacian) induces the "radial" Laplacian on $T$:*

$$L_r = \Delta_T + \sum_{\alpha \in R^+} m_\alpha \cot(\alpha \cdot h) \frac{\partial}{\partial \alpha} \tag{17}$$

*where $\Delta_T$ is the standard Laplacian on the torus. Here, $h$ is the "flat" coordinate of $x$ on the maximal torus (e.g. for sphere this is the angle between $x$ and $x_0$).*

One can readily see that the above formula allows one to reduce computations over $\mathcal{M}$ to computations over $T$, which greatly reduces dimensionality ($n$ dimensions to 1 dimension for spheres and $O(n^2)$ dimensions to $O(n)$ for Matrix Lie Groups).

## 3.2 Improved Heat Kernel Estimation

We now use this maximal torus perspective to improve computations related to the heat kernel. In particular, we can greatly improve both the speed and fidelity of the numerical evaluation during our training process.

### 3.2.1 Eigenfunction Expansion Restricted to the Maximal Torus

We note that the maximal torus relationship in Proposition 3.4 reduces the eigenfunction expansion in Equation 13 to an eigenfunction expansion of the induced Laplacian on the maximal torus. This has implicitly appeared in previous work when defining Riemannian Diffusion Models on $S^2$ and $SO(3)$ [4, 34], allowing one to rewrite the summation as, respectively

$$K_{S^2}^{EF}(x|x_0, t) = \frac{1}{4\pi} \sum_{l=0}^{\infty} (2l+1) P_l(\langle x, x_0 \rangle) e^{-l(l+1)t} \tag{18}$$

where $P_l$ are the Legendre Polynomials and

$$K_{SO(3)}^{EF}(x|x_0, t) = \frac{1}{8\pi^2} \sum_{l=0}^{\infty} e^{-2l(l+1)t} \frac{\sin(2l+1)\theta/2}{\sin(\theta/2)} \quad \theta \text{ is the angle between } x, x_0 \tag{19}$$

By making this relationship explicit, we can directly generalize this formulation to other symmetric spaces, e.g. $SU(3)$

$$K_{SU(3)}^{EF}(x|x_0, t) = \sum_{p,q=1}^{\infty} \frac{pq(p+q)}{4} (\chi^{p,q}(\theta, \phi) + \chi^{q,p}(\theta, \phi)) e^{-\frac{t(p^2+q^2+pq)}{6}} \tag{20}$$

where $\chi$ are the (real) irreducible representations and $\theta, \phi$ are the induced angles between $x$ and $x_0$.

By reducing the dimensionality, this strategy generally makes the summation more tractable. Furthermore, for cases such as the hypersphere $S^n$, the eigenfunctions are numerically unstable (rendering them useless for our purposes), while the maximal torus setup stable.

### 3.2.2 Controlling Errors for Small Time Values

While the torus eigenfunction expansion greatly reduces the computational cost (particularly for higher dimensions), for small values of $t$ the heat kernel scaling terms $e^{-\lambda_i t}$ remain large as one increases $i, \lambda_i \to \infty$. As a result, one must evaluate many more terms (often in the thousands), and the result can be very numerically unstable even with double precision.

To address this problem, we examine several refined versions of the Varadhan approximation that use the fact that the manifold is symmetric. These approximations allow us to control the number of eigenfunctions required and, in some cases, completely obviate the need for them entirely.

**The Schwinger-Dewitt Approximation**

The Varadhan approximation is constructed by approximating the heat kernel with a Gaussian distribution (with respect to the Riemannian distance function). However, doing this does not account for the curvature of the manifold. Incorporating this premise, we get the Schwinger-Dewitt approximation [9]:

$$K_{\mathcal{M}}^{\text{SD}}(x|x_0, t) = \frac{\overline{\Delta}_{x_0}(x)^{1/2} e^{-\frac{d_{\mathcal{M}}(x_0, x)^2}{4t} + \frac{tR}{6}}}{(4\pi t)^{\dim \mathcal{M}/2}} \tag{21}$$

Here, $\overline{\Delta}_{x_0}(x) = \det(D_{x_0} \exp_{x_0}(\log_{x_0}(x)))$ is the (unnormalized) change of volume term introduced by the exponential map, and $R$ is the scalar curvature of the manifold. Generally, this is much more accurate than Varadhan's approximation as it better accounts for the curvature of the manifold through the $\Delta$ term, retaining accuracy up to moderate time values.

$\Delta$ appears to be a rather computationally demanding term since it requires evaluating the determinant of a Jacobian. Indeed, this naively takes $O(\dim \mathcal{M}^3)$ evaluations which is completely inaccessible in higher dimensions. However, we again emphasize the fact that we are working with symmetric spaces: $\Delta$ has a particularly simple formula defined by our flat coordinate $h$ from above:

$$\overline{\Delta}_{x_0}(x) = \prod_{\alpha \in R^+} \left( \frac{\alpha \cdot h}{\sin(\alpha \cdot h)} \right)^{m_\alpha} \tag{22}$$

**Sum Over Paths**

The fact that we can derive a better approximation using a different power of $\Delta$ points to deeper connections between the heat kernel and the "Gaussian" with respect to distance. We draw inspiration from several Euclidean case examples, such as the flat torus [28] or the unit interval [36]. For those cases, the heat kernel is derived by summing a Gaussian over all possible paths connecting $x_0$ and $x$. While this formula does not exactly lift over to Riemannian symmetric spaces, there exists an analogue for Lie Groups [9]:

$$K_{\mathcal{M}}^{\text{SOP}}(x|x_0, t) = \frac{e^{t\rho^2}}{(4\pi t)^{\dim \mathcal{M}/2}} \sum_{2\pi n \in \Gamma} \prod_{\alpha \in R^+} \frac{\alpha \cdot (h + 2\pi n)}{2\sin(\alpha \cdot h/2)} e^{-\frac{(h + 2\pi n)^2}{4t}} \tag{23}$$

Here, $\Gamma$ is infinite tangent space grid that corresponds with loops of the torus (i.e. $2\pi n$ circular intervals), and $\rho^2$ is a manifold specific constant. We note that the product over $R^+$ is exactly the $\Delta$ change in variables as given above since $m_\alpha = 1$, but we simply extend this to every other root.

Generally, this formula is rather powerful as it gives us an exact (albeit infinite) representation for the heat kernel. Compared to the eigenfunction expansion in Equation 13, the Sum Over Paths representation is accurate for small $t$, which nicely complements the fact that the eigenfunction representation is accurate for large $t$. This formula does generalize to split-rank Riemannian symmetric spaces like odd dimensional spheres (more details in Appendix A.1) and noncompact spaces (Appendix A.2).

### 3.2.3 A Unified Heat Kernel Estimator

We unify these approximations into a single heat kernel estimator. Our computation method splits up the heat kernel evaluation based on the value of $t$ and applies an eigenfunction summation or an improved small time approximation accordingly. This allows us to effectively control the errors at both the small and large time steps while significantly reducing the number of function evaluations for each. Our full algorithm is outlined in Algorithm 1.

**Algorithm 1:** Heat Kernel Computation

---

**Hyperparameters:** Riemannian symmetric space $\mathcal{M}$, number of eigenfunctions $n_e$, time value
cutoff $\tau$, (optional, depending on if $\mathcal{M}$ is a Lie Group) number of paths $n_p$
**Input:** source $x_0$, time $t$, query value $x$
Compute
**if** $t < \tau$ **then**
    **if** $\mathcal{M}$ is a Lie Group **then**
        |  **return** $K_{\mathcal{M}}^{\text{SOP}}(x|x_0, t)$ truncated to $|n| < n_p$ in the summation over $\Gamma$.
    **else**
        |  **return** $K_{\mathcal{M}}^{\text{SD}}(x|x_0, t)$.
    **end**
**else**
    |  **return** $K_{\mathcal{M}}^{\text{EF}}(x|x_0, t)$ truncated to $|n| < n_e$.
**end**
**remark** $\nabla_x \log K_{\mathcal{M}}$ can be computed with autodifferentiation.

---

We ablate the accuracy of our various heat kernel (score) approximations in Figure 1 for $S^2$, $S^{127}$,
and $SO(3)$. In general, we found that computing with standard eigenfunctions was too costly and
too prone to numerical blowup, and Varadhan's approximation was simply too inaccurate. For some
spaces like $S^{127}$, a combination of preexisting methods (like is briefly explored in [4]) would not
work since $K_{\mathcal{M}}^{\text{EF}}$ NaNs out before Varadhan becomes accurate.

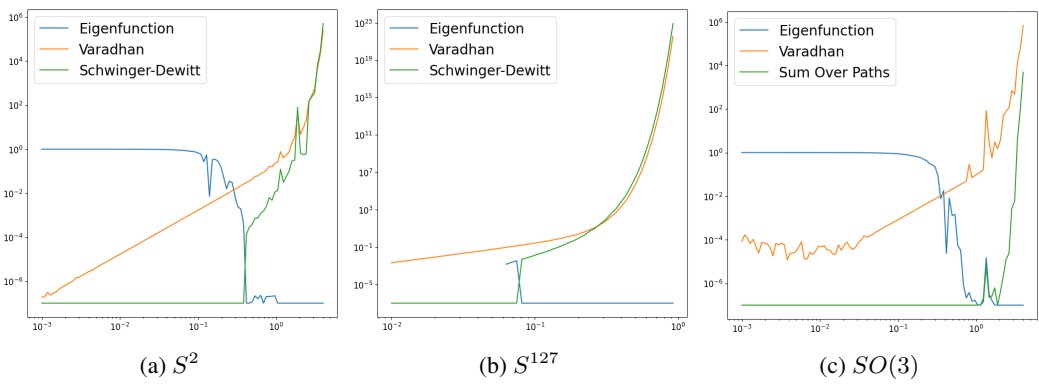

(a) $S^2$             (b) $S^{127}$             (c) $SO(3)$

Figure 1: **We compare the various heat kernel estimators on a variety of manifolds.** We plot
the relative error compared with $t$. Our improved small time asymptotics allow us to control the
numerical error, while baseline Varadhan is insufficient. For $S^{127}$, Varadhan will still produce an
error of 10% when the eigenfunction expansion NaNs out, so we need to use Schwinger-Dewitt

### 3.3 Exact Heat Kernel Sampling

To train with the denoising score matching objective, one must produce heat kernel samples to
monte carlo samples the loss. Typically, Riemannian Diffusion Models sample by discretizing a
Brownian motion through a geodesic random walk [29]. However, this can be slow as it requires
taking many computationally expensive exponential map steps on $\mathcal{M}$ and can drift off the manifold
due to compounding numerical error [25]. In this section, we discuss strategies to sample from the
heat kernel quickly and exactly.

**Cheap Rejection Sampling Methods**

When our dimension is low enough, we can sample using rejection sampling using our fast heat kernel
evaluator. The key detail is the prior distribution, which needs to have a closed form density, be easy
to sample from, and must not deviate from the heat kernel too much. For large time steps $t$ the natural
prior distribution is uniform. Conversely, for small time steps, we instead use the wrapped Gaussian
distribution, which can be sampled by passing a tangent space Gaussian through the exponential map

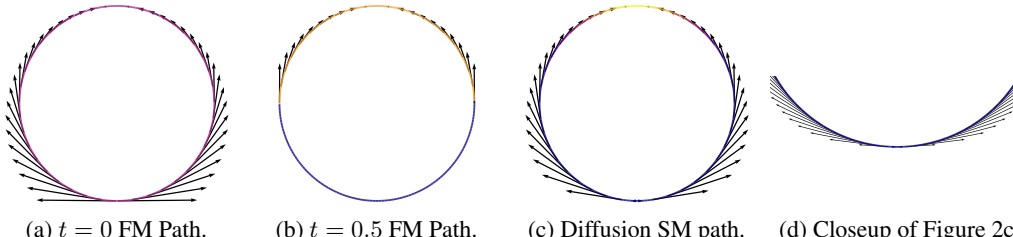

(a) $t = 0$ FM Path.    (b) $t = 0.5$ FM Path.    (c) Diffusion SM path.    (d) Closeup of Figure 2c

Figure 2: **We visualize the vector fields generated by the flow matching geodesic path and our score matching diffusion path.** These are done on $S^1$. (a) The flow matching path has a discontinuity at the pole. (b) The marginal densities of the flow matching path are not smooth and transition sharply at the boundary. (c) Our score matching path has a smooth density and smooth vectors. (d) At the pole, our score matching path anneals to $0$ to maintain continuity.

and has a density

$$p_{\text{wrap}}(x|x_0, t) = \frac{1}{(4\pi t)^{\dim \mathcal{M}/2}} \sum_{2\pi n \in \Gamma} \prod_{\alpha \in R^+} \left( \frac{\sin(\alpha \cdot h)}{\alpha \cdot (h + 2\pi n)} \right) e^{-\frac{(h + 2\pi n)^2}{4t}} \tag{24}$$

In both cases, the ratio can be trivially bounded by examining their behavior numerically.

**Heat Kernel ODE Sampling**

As an alternative, we notice that we can apply the probability flow ODE to sample from the heat kernel exactly. In particular, we draw a sample $x_T \sim \mathcal{U}_{\mathcal{M}}$, where $T$ is large enough s.t. $K_{\mathcal{M}}(\cdot|x_0, T)$ is close to the uniform distribution (within numerical precision). We then solve the ODE $\frac{d}{ds}x_T = -\frac{1}{2}\nabla_x \log K_{\mathcal{M}}(x_s|x_0, s)$ from $s = T$ to $s = t$. As the sampling procedure follows the probability flow ODE, this is guaranteed to produce samples from $K_{\mathcal{M}}(\cdot|x_0, t)$.

This is solvable as a manifold ODE, as previous works have already developed adaptive manifold ODE solvers [37]. Furthermore, by Proposition 3.4, we can restrict our vector field to the maximal torus and solve it there. Note that this allows us to use preexisting Euclidean space solvers since the torus is effectively Euclidean space. Lastly, we can scale the time schedule of the ODE (with a scheme like a variance-exploding schedule) to stabilize the numerical values.

## 4 Related Work

Our work exists in the established framework of differential equation-based Riemannian generative models. Early methods generalized Neural ODEs to manifolds[15, 37, 38], enabling training with maximal likelihood. More recent methods attempt to remove the simulation components[3, 42], but this results in unscalable or biased objectives. We instead work with diffusion models, which are based on scores and SDEs and do not have any of these issues. In particular, we aim to resolve the main gap that prevents Riemannian Diffusion Models from scaling to high dimensions.

Riemannian Flow Matching (RFM) [10] is a concurrent work that attempts to achieve similar goals (i.e. scaling to high dimensions) by generalizing flow matching [35] to Riemannian manifolds. The fundamental difficulty is that one must design smooth vector fields that flows from a base distribution (ie the uniform distribution) to a data point. RFM introduces several geodesic-based vector fields, but these break the smoothness assumption (see Figure 2), which we found to be detrimental for some of the densities we considered (e.g. Figure 3). However, similar to the Euclidean case, RFM with the diffusion path corresponds to score matching with the probability flow ODE, so our work provides the numerical computation necessary to construct such a flow.

## 5 Experiments

### 5.1 Simple Test Tasks

We start by comparing our accurate denoising score matching objective with the inaccurate approximation scheme suggested by [4] based on $50$ eigenfunctions. We test on the compiled Earth science

| Method ($S^2$) | Volcano | Earthquake | Flood | Fire |
|---|---|---|---|---|
| Sliced Score Matching | -4.92 $_{\pm\,0.25}$ | -0.19 $_{\pm\,0.07}$ | 0.45 $_{\pm\,0.17}$ | -1.33 $_{\pm\,0.06}$ |
| Denoising Score Matching (inaccurate) | -1.28 $_{\pm\,0.28}$ | 0.13 $_{\pm\,0.03}$ | 0.73 $_{\pm\,0.04}$ | -0.60 $_{\pm\,0.18}$ |
| Denoising Score Matching (accurate) | -4.69 $_{\pm\,0.29}$ | -0.27 $_{\pm\,0.05}$ | 0.44 $_{\pm\,0.03}$ | -1.51 $_{\pm\,0.13}$ |

Table 1: **We measure the improvement of our improved heat kernel estimator on downstream climate science tasks and report negative log likelihood (↓).** Without our accurate heat kernel estimator, the denoising score matching loss produces substandard results.

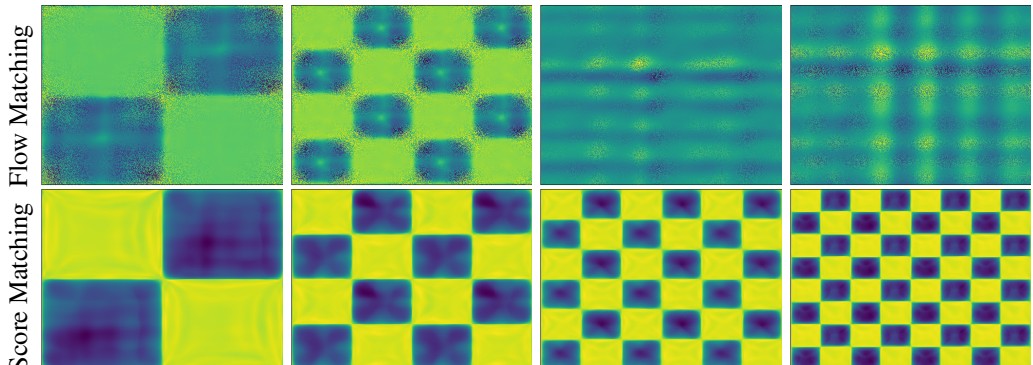

Figure 3: **We compare Riemannian score matching and flow matching on increasingly complex checkerboard patterns on the torus.** Flow matching learns suboptimal distributions with noticeable artifacts (like blurriness and spurious peaks) for simpler distributions and fails outright for more complex checkerboards. Conversely, Riemannian score matching learns accurate densities.

datasets from [38], detailing results are in Table 1. Generally, our accurate heat kernel results in a substantial improvement and matches sliced score matching. Note that we do not expect our method to outperform sliced score matching since these datasets are low dimensional and the objectives are equivalent (up to a small additional variance).

We also compare directly with RFM on a series of increasingly complex checkerboard datasets on the flat torus. These datasets have appeared in prior work to measure model quality [3, 5, 37]. As a result of the non-smooth vector field dynamics, we find that RFM degrades in performance as the checkerboard increases in complexity, and is unable to learn past a certain point. Our visualized results are given in Figure 3.

## 5.2 Learning the Wilson Action on $SU(3)$ Lattices.

We apply our method to learn densities on $SU(3)^{4\times4}$. We consider the data density defined by the Wilson Action [53]:

$$p(K) \propto e^{-S(K)} \quad S(K) = -\frac{\beta}{3} \sum_{x,y\in\mathbb{Z}_4^2} \operatorname{Re} \operatorname{tr}(K_{x,y}K_{x+1,y}K^*_{x+1,y+1}K^*_{x,y+1}) \tag{25}$$

For our experiments, we take $\beta = 9$ as a standard hyperparameter and learn this distribution, achieving an effective sample size of 0.62. Generally, learning to sample from $p(K)$ directly requires a separate objective that is not based on score matching[6, 50]. However, to showcase our score matching techniques, we instead learn the density through precomputed samples, and concurrent work has shown that this type of training can be coupled with diffusion guidance to improve variational inference methods [17]. As such, our model has the potential to improve $SU(n)$ lattice QCD samplers, although we leave this task for future work. We also note that our model can likely be further improved by building data symmetries into the score network [30].

## 5.3 Contrastively Learned Hyperspherical Embeddings

Finally, we examine contrastively learning [12]. Standard contrastive losses optimize embeddings of the data that lie on the hypersphere. Paradoxically, these embeddings are unsuitable for most

| Method | SVHN | Places365 | LSUN | iSUN | Texture |
|---|---|---|---|---|---|
| SSD+[44] | 31.19 | 77.74 | 79.39 | 80.85 | 66.63 |
| KNN+[49] | 39.23 | 80.74 | 48.99 | 74.99 | 57.15 |
| CIDER (penultimate layer) | 23.09 | 79.63 | 16.16 | 71.68 | 43.87 |
| CIDER (hypersphere) | 93.53 | 89.92 | 93.68 | 89.92 | 92.41 |
| CIDER (hypersphere) + Diffusion | 28.95 | 76.54 | 35.78 | 74.17 | 62.87 |

Table 2: **We compare contrastive learning OOD detection methods on CIFAR-100.** We report false positive rates (↓) for $0.05$ false negative rate. The hyperspherical embeddings produce very bad results, but with a Riemannian Diffusion Model, it is competitive with or surpass the state of the art.

downstream tasks, so practitioners instead use the penultimate layer [2]. This degrades interpretability, since the theoretical analyses in this field work on the hyperspherical embeddings [52].

We investigate this issue further in the context of out of distribution (OOD) detection. We use the pretrained embedding network from CIDER[39]. Using the hyperspherical representation for OOD detection produces very subpar results. However, using likelihoods from our Riemannian Diffusion Model stabilizes performance and achieves comparable results with other penultimate feature-based methods (see Figure 2). We emphasize that the embedding network has been tuned to optimize the performance using the penultimate layer. Since our established theory exclusively focuses on the properties of the hyperspherical embedding, the fact that our Riemannian Diffusion Models can extract a comparable representation can lead to more principled improvements for future work.

## 6 Conclusion

We have introduced several practical improvements for Riemannian Diffusion Models that leverage the fact that most relevant manifolds are Riemannian symmetric spaces. Our improved capabilities allow us, for the first time, to scale differential equation manifold models to hundreds of dimensions, where we showcase applications in lattice QCD and constrastive learning. We hope that our improvements help open the door to the broader adoption of Riemannian generative modeling techniques.

## 7 Acknowledgements

This project was supported by NSF (#1651565), ARO (W911NF-21-1-0125), ONR (N00014-23-1-2159), and CZ Biohub. AL is supported by a NSF Graduate Research Fellowship and MX is supported by a Stanford Graduate Fellowship.

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

# A Additional Heat Kernel Computational Details

## A.1 Sum over Paths for Spheres

For spheres, one has the following sum over paths formula for odd-dimensional spheres $S^n$:

$$K_{S^n}(y|x,t) = e^{-(n-1)^2 t/2} \frac{1}{2\pi} \left( -\frac{1}{\sin\theta} \frac{\partial}{\partial\theta} \right)^{\frac{n-1}{2}} K_{S^1}(y|x,t) \tag{26}$$

where $\theta = \arccos(x \cdot y)$ is the distance between two points. One can derive the sum over paths formulation by applying the derivative to each summand. This formula does not generalize to even dimensional spheres, as the intertwining operator appears which makes the evaluation an integral, although a similar recurrent can be derived.

## A.2 Extension to Noncompact Symmetric Spaces

For noncompact symmetric spaces, our heat kernel evaluations are only given by the sum over paths representation. Luckily, there is only "one" path since the space is not compact, meaning we don't need to do an infinite sum. In particular, this gives us the following formula for noncompact lie groups

$$K_G(y|x,t) = \frac{e^{t\rho^2}}{(4\pi t)^{\dim \mathcal{M}/2}} \prod_{\alpha \in R^+} \frac{\alpha \cdot (h + 2\pi n)}{2\sinh(\alpha \cdot h/2)} e^{-\frac{h^2}{4t}} \tag{27}$$

and for hyperbolic space, one has the formula for odd dimensions (equivalent to the one for odd dimensional spheres):

$$K_{H^n}(y|x,t) = \left( -\frac{1}{2\pi} \right)^{\frac{n-1}{2}} \frac{1}{2\sqrt{\pi t}} \left( \frac{1}{\sinh\rho} \frac{\partial}{\partial\rho} \right)^{\frac{n-1}{2}} e^{-\frac{(n-1)^2}{4}t - \frac{\rho^2}{4t}} \tag{28}$$

where $\rho$ is the hyperbolic distance between $y$ and $x$.

Generally, there is the notion of a OU process on a manifold [4, 25] which limits towards the wrapped Gaussian distribution. We note that the intermediate densities are not heat kernels (unlike in the Euclidean case).

## A.3 Explicit Heat Kernel Formulas

In this section, we highlight the formulas we used for computing the heat kernel.

**Torus.** The Torus $T^n \cong (S^1)^n$ can be realized as a flat torus $[0, 2\pi)^n$, where each coordinate represents the angular component. Under this construction, we can compute the kernel for each coordinate in $S^1 \cong [0, 2\pi)$ and then take the product. The eigenfunction expansion is

$$K_{S^1}(y|x,t) = \frac{1}{\pi} \left( \frac{1}{2} + \sum_{k=1}^{\infty} e^{-k^2 t} (\cos(kx)\cos(ky) + \sin(kx)\sin(ky)) \right) \tag{29}$$

The heat kernel also admits a sum over paths representation. In particular, this agrees with the wrapped probability since the change of volume term is 1:

$$K_{S^1}(y|x,t) = \frac{1}{\sqrt{4\pi t}} \sum_{k=-\infty}^{\infty} e^{-\frac{(y-x+2\pi k)^2}{4t}} \tag{30}$$

**Spheres.** The sphere $S^n$ is the set $\{x \in \mathbb{R}^{n+1} : \|x\| = 1\}$. We use the formulas (eigenfunction and Schwinger-Dewitt) given in the main text, noting that the maximal torus value can be extracted with the equation $\theta = \arccos(x \cdot y)$ (ie the geodesic distance).

**SO(3)**. $SO(3)$ is the Lie Group $\{X \in \mathbb{R}^{3\times3} : XX^\top = I, \det(X) = 1\}$. We have already given the eigenfunction expansion in the main text, but the sum over paths method can be derived from the fact that the maximal torus value $\theta$ is the distance between $x$ and $y$ on the manifold. The sum over paths formula is given by

$$K_{SO(3)}^{\text{SOP}}(y|x,t) = \frac{e^{\frac{t}{16}}}{(2\pi t)^{\frac{3}{2}}} \sum_{n=-\infty}^{\infty} \frac{\theta + 2\pi n}{\sin(\theta)} e^{-(\theta+2\pi n)/t} \tag{31}$$

**SU(3)**. $SU(3)$ is the (real) Lie Group $\{X \in \mathbb{C}^{3\times3} : XX^H = I, \det(X) = 1\}$. The eigenfunction expansion can be derived from the character classes [16], and the sum over paths representation can be derived directly [1] as $K_{SU(3)}(y|x,t)^{\text{SOP}} =$

$$\frac{1}{S} \sum_{l=-\infty}^{+\infty} \sum_{m=-\infty}^{\infty} (A(l) - B(m))(A(l) + 2B(m))(2A(l) + B(m)) e^{-\frac{A(l)^2 + B(m)^2 + A(l)B(m)}{t}} \tag{32}$$

where $A, B$ are the maximal torus values of $x^{-1}y$ (these are $\log(\lambda)/i$ for eigenvalues $\lambda$ of $x^{-1}y$), $S = 8 \sin \frac{A-B}{2} \sin \frac{2A+B}{2} \frac{A+2B}{2}$

## B Experimental Details

### B.1 Heat Kernel Estimates

We sample a random point (based off of the heat kernel probability) for each time step to compute.

$\mathbf{S^2}$. We use 10000 eigenfunctions for the ground truth and 10 for our comparison.

$\mathbf{S^{127}}$. We use 50000 eigenfunctions evaluated at double precision for our ground truth and 100 for our comparison.

$\mathbf{SO(3)}$. We use 10000 eigenfunctions for our ground truth and 50 for our comparison. We sum over 10 paths.

### B.2 Earth Science Datasets

We do not perform a full hyperparameter search. We use a very similar architecture to the one used in Bortoli et al. [4] except we use the SiLU activation function without a learnable parameter [23] and a learning rate of $5 \cdot 10^{-4}$.

### B.3 2D Torus

We use a standard MLP with 4 hidden layers and the SiLU activation function and learn with the Adam optimizer with learning rate set to $1e-3$ [31]. However, we transform the input $x \to \sin(kx), \cos(kx)$ where $k$ ranges from 1 to 6. This was done to ensure that the input respects the boundary constraints. We note that this architecture is generally quite powerful, as the Fourier coefficients can capture finer grain features, but this was optimized for the flow matching baseline, not the score matching method. In particular, score matching works with significantly fewer/no Fourier coefficients. We train for 100000 gradient updates with a batch size of 100 (each batch is randomly sampled from the checkerboard).

### B.4 SU(3) Lattice

We generate our 20000 ground truth samples using Riemannian Langevin dynamics with a step size of $1e-3$ for 10000 update iterations. Our model is similar to the model used in Kingma et al. [32], except we circular pad the convnet and use 3 layers for each up-down block instead. We input a compressed version of the $3 \times 3$ $SU(n)$ matrix, making the input size 18. We train with a learning rate of $5 \cdot 10^{-4}$ and perform 1000000 updates with a batch size of 512. To evaluate, we use an 0.999 exponential moving average [40] and sample using the manifold ODE sampler [37].

## B.5  Contrastive Learning

We use the pretrained networks given by Ming et al. [39] to construct our hyperspherical embeddings. Our Riemannian diffusion model is similar to the simplex diffusion model given by [36], although we use 3 layers instead of 4. We train using the Adam optimizer with a learning rate of $5 \cdot 10^{-4}$, performing a 0.999 EMA before using the manifold ODE solver to evaluate likelihoods.

