# Scaling Riemannian Diffusion Models

## Abstract

Riemannian diffusion models draw inspiration from standard Euclidean space diffusion models to learn distributions on general manifolds. Unfortunately, the additional geometric complexity renders the diffusion transition term inexpressible in closed form, so prior methods resort to imprecise approximations of the score matching training objective that degrade performance and preclude applications in high dimensions. In this work, we reexamine these approximations and propose several practical improvements. Our key observation is that most relevant manifolds are symmetric spaces, which are much more amenable to computation. By leveraging and combining various ansätze, we can quickly compute relevant quantities to high precision. On low dimensional datasets, our correction produces a noticeable improvement and is competitive with other techniques. Additionally, we show that our method enables us to scale to high dimensional tasks on nontrivial manifolds, including $SU(n)$ lattices in the context of lattice quantum chromodynamics (QCD). Finally, we apply our models to contrastively learned hyperspherical embeddings, curbing the representation collapse problem in the projection head and closing the gap between theory and practice.

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

$$d\mathbf{x}_t = d\mathbf{B}_t \tag{11}$$

which can easily be rescaled by time $\int_0^t g(t)ds$ given a time schedule $g(t)$ (though this is omitted for clarity purposes). Unlike the Euclidean case, here $p_t$ is stationary and approaches the uniform distribution $\mathcal{U}_\mathcal{M}$ as $t \to \infty$ (in practice, this convergence is fast, getting within numerical precision for $t \approx 5$).

**The transition density has no closed form.** Despite the fact that we work with the most simple SDE, the transition kernel defined by the manifold heat equation $p_t(x_t|x_0)$ has no closed form. This transition kernel is known as the heat kernel, which satisfies Equation 2 with the additional condition that, as $t \to 0$, the kernel approaches $\delta_{x_0}$. We will denote this by $K_\mathcal{M}(x_t|x_0, t)$, and we highlight that, when $\mathcal{M}$ is $\mathbb{R}^d$, this corresponds to a Gaussian and is easy to work with.

This has several major consequences which cause prior to work to favor sliced score matching over denoising score matching. First, to sample a point $x \sim K_\mathcal{M}(\cdot|x_0, t)$, one must simulate a Geodesic Random Walk

$$x_{t+\Delta t} = \exp_x(\sqrt{\Delta t}z) \quad z \sim \mathcal{N}(0, I_d) \in T_x\mathcal{M} \tag{12}$$

where $\exp$ is the Riemannian exponential map. Additionally, to calculate $K_\mathcal{M}(\cdot|x_0, t)$ or $\nabla \log K_\mathcal{M}(x|x_0, t)$, one must truncate the eigenfunction representation

$$K_\mathcal{M}^{\text{EF}}(x|x_0, t) = \sum_{i=0}^{\infty} e^{-\lambda_i t} f_i(x_0) f_i(x) \tag{13}$$

Here, $f_i, \lambda_i$ are the discrete eigenfunctions/eigenvalues of the Laplacian $\Delta f_i = -\lambda_i f_i$ and form an orthonormal basis for all $L^2$ functions on $\mathcal{M}$. Previous work has also explored the use of the Varadhan approximation for small values of $t$ (which uses the Riemannian logarithmic map) [44]:

$$K_\mathcal{M}(x|x_0, t) \approx \mathcal{N}(0, \sqrt{2t})(\text{dist}_\mathcal{M}(x_0, x)) \implies \nabla_x \log K_\mathcal{M}(x|x_0, t) \approx \frac{1}{2t} \log_x(x_0) \tag{14}$$

## 3 Method

The key problem with applying denoising score matching in practice is that the heat kernel computation is too expensive and inaccurate. For example, simulating a geodesic random walk is expensive since it requires many exponential maps. Furthermore, the eigenfunction expansion in Equation 13 requires increasingly more eigenfunctions as $t \to 0$ (numbering in the tens of thousands). Worse still, their formulas are not well-known for most manifolds, and, even when explicit formulas exist, they can be numerically unstable (like in the case of $S^n$). One possible way to alleviate this is to use Varadhan's approximation for small $t$, but this is also unreliable except for very small $t$.

To remedy this issue, we instead consider the case of Riemannian symmetric spaces [22]. We emphasize that most manifold generative modeling applications already model on Riemannian Symmetric Spaces like the sphere, torus, or Lie Groups, so we do not lose applicability by restricting our attention here. Furthermore, for surfaces (which have appeared as generative modeling test tasks in the literature [43]), one can always define a generative model by mapping the data points to $S^2$, learning a generative model there, and mapping back [14]. We empathize that, outside of these two examples, we are unaware of any other manifolds which have been used for Riemannian generative modeling tasks.

### 3.1 Heat Kernels on Riemannian Symmetric Spaces

In this section, we will define Riemannian Symmetric Spaces and showcase relationships with the heat kernel. We empathize that our exposition is neither rigorous nor fully defines all terms. We urge interested readers to consult a book [22] or monograph [9] for a full treatment of the subject.

**Definition 3.1.** *A Riemannian Symmetric Space is a Riemannian manifold such that, for all points $x \in \mathcal{M}$, there exists a local isometry $s$ s.t. $s(x) = x$ and $D_x s = -\mathrm{id}_{T_x \mathcal{M}}$.*

This symmetry property is relatively simple but has numerous ramifications. In particular, we can characterize all Riemannian symmetric spaces as a quotient $G/K$ where $G$ is a Lie Group and $K$ is a compact isotropy group.

**Examples.** *This includes many well known manifolds, such as Lie Groups $G \cong G/\{e\}$ (where $\{e\}$ is the trivial Lie Group), the sphere $S^n \cong SO(n+1)/SO(n)$, and hyperbolic space $H^n \cong SO(n,1)/O(n)$.*

In our paper, we do not consider the case of noncompact Riemannian symmetric spaces, as these are diffeomorphic to Euclidean space. As such, we can reapply the same generative modeling trick that we used for surfaces: map data points to $\mathbb{R}^n$, learn a standard diffusion model there, and map back.

On symmetric spaces, one can define a special structure called the maximal torus which is critical for our analysis. Intuitively, the maximal torus parameterizes the symmetries of the space.

**Definition 3.2** (Maximal Torus). *A torus on a Lie group is any compact, connected, and abelian subgroup of $G$. These are isomorphic to standard tori $T^m \cong (S^1)^m$. A maximal torus $T$ is a torus which is not contained in any other torus. All maximal tori are conjugate. Symmetric spaces inherit maximal tori from their quotient space $G$.*

**Examples.** *For the Lie group of unitary matrix $U(n)$, the maximal torus is defined as $T = \{\mathrm{diag}(e^{i\theta_1}, \ldots, e^{i\theta_n}) : \theta_k \in [0, 2\pi)\}$. For the sphere $S^n$, a maximal torus is any great circle.*

We write $R^+$ as the set of all the positive roots of our symmetric spaces (these are values on the maximal torus), and for a root $\alpha$ we let $m_\alpha$ be the multiplicity (e.g. for spheres, $\alpha$ is 1 and $m_\alpha$ is $d-1$). Furthermore, for a point $x$, we let $h$ be the "flat" coordinates of $x$ on the maximal torus (e.g. for spheres $h$ is the angle between $x$ and an anchor point $x_0$). Then, we can rewrite the heat kernel in terms of the maximal torus:

**Proposition 3.3** (Heat Kernel Reduces on Maximal Torus). *The Laplace-Beltrami operator on $\mathcal{M}$ (the manifold generalization of the standard Laplacian) induces the "radial" Laplacian on $T$:*

$$L_r = \Delta_T + \sum_{\alpha \in R^+} m_\alpha \cot(\alpha \cdot h) \frac{\partial}{\partial \alpha} \tag{15}$$

*where $\Delta_T$ is the standard Laplacian on the torus. As such, the heat kernel reduces to a function of $h$.*

## 3.2 Improved Heat Kernel Estimation

We now use the fact that the heat kernel is intimately connected with the maximal torus to better estimate the heat kernel values. This greatly improves the speed and fidelity of our numerical evaluation during training.

### 3.2.1 Eigenfunction Expansion Restricted to the Maximal Torus

We note that the maximal torus relationship in Proposition 3.3 reduces the eigenfunction expansion in Equation 13 to an eigenfunction expansion of the induced Laplacian on the maximal torus. This has implicitly appeared in previous work when defining Riemannian Diffusion Models on $S^2$ and $SO(3)$ [4, 34], allowing one to rewrite the summation as, respectively

$$K_{S^2}(x|x_0, t) = \frac{1}{4\pi} \sum_{l=0}^{\infty} (2l + 1) P_l(\langle x, x_0 \rangle) e^{-l(l+1)t} \tag{16}$$

where $P_l$ are the Legendre Polynomials and

$$K_{SO(3)}(x|x_0, t) = \frac{1}{8\pi^2} \sum_{l=0}^{\infty} e^{-2l(l+1)t} \frac{\sin(2l+1)\theta/2}{\sin(\theta/2)} \quad \theta \text{ is the angle between } x, x_0 \tag{17}$$

By making this relationship explicit, we can draw upon similar formulas for other symmetric spaces, e.g. the hypersphere [39]

$$K_{S^n}(x|x_0, t) = \frac{1}{V(S^n)} \sum_{l=0}^{\infty} \frac{2l + n - 1}{n - 1} G_l^{(n-1)/2}(\langle x, x_0 \rangle) e^{-l(l+d-1)t} \tag{18}$$

where $G_l^\alpha$ are the Gegenbauer polynomials and $V(S^n)$ is the volume of $S^n$. The new summation works by effectively "collapsing" the summation for eigenfunctions $f_i$ with the same eigenvalue $\lambda_i$. This drastically reduces the number of computations required from $O(M^{\dim \mathcal{M}})$ to $O(M^{\dim T})$, where $M$ is the cutoff value. Furthermore, this also tends to greatly simplify the explicit formula. As an example, for $S^n$ this reduces the computation from $O(M^n)$ to $O(M)$ (since we no longer need to evaluate $O(i)$ eigenfunctions for each eigenvalue $\lambda_i$) and avoids the computation of numerically unstable hyperspherical harmonics.

### 3.2.2 Controlling Small Time Errors

While the torus eigenfunction representation greatly reduces the computational cost (particularly for several higher-dimensional manifolds), they still require thousands of eigenfunctions for small values of $t$. Worse still, numerical error persists: for small values of $t$, computing the eigenfunction expansion can easily cause overflow errors that even double precision can't resolve. To this end, we examine several refined versions of the Varadhan approximation that use the fact that the manifold is a Riemannian symmetric space. These approximations can allow us to control the number of eigenfunctions required and, in some cases, completely obviate the need for them altogether.

**The Schwinger-Dewitt Approximation**

The Varadhan approximation is built by approximating the heat kernel with a Gaussian distribution with respect to the Riemannian distance function. However, doing this does not account for the curvature of the manifold. By accounting for this curvature, we derive the Schwinger-Dewitt approximation [9]:

$$K_{\mathcal{M}}^{\text{SD}}(x|x_0, t) = \frac{\overline{\Delta}_{x_0}(x)^{1/2} e^{-\frac{d_{\mathcal{M}}(x_0, x)^2}{4t} + \frac{tR}{6}}}{(4\pi t)^{\dim \mathcal{M}/2}} \tag{19}$$

Here $\overline{\Delta}_{x_0}(x) = \det(D_{x_0} \exp_{x_0}(\log_{x_0}(x)))$ is the (unnormalized) change of volume term introduced by the exponential map, and $R$ is the scalar curvature of the manifold. Generally, this is much more stable than Varadhan's approximation as it better accounts for the curvature of the manifold, retaining accuracy up to moderate time values.

$\Delta$ appears to be a rather computationally demanding term. Indeed, naïve calculations require the formation of the full Jacobian matrix and a determinant computation, which scales poorly with

dimensions and is completely inaccessible in higher dimensions. However, we again emphasize the
fact that we are working with symmetric spaces; here, $\Delta$ has a particularly simple formula defined by
our flat coordinate $h$ from above:

$$\overline{\Delta}_{x_0}(x) = \prod_{\alpha \in R^+} \left( \frac{\alpha \cdot h}{\sin(\alpha \cdot h)} \right)^{m_\alpha} \tag{20}$$

**Sum Over Paths**

The fact that we can derive a better approximation using a different power of $\Delta$ points to deeper
connections between the heat kernel and the "Gaussian" with respect to distance. We draw inspiration
from several Euclidean case examples, such as the flat torus [28] or the unit interval [36]. For these
cases, the heat kernel is derived by summing a Gaussian over all possible paths connecting $x_0$ and $x$.
While this formula does not exactly lift over to Riemannian symmetric spaces, there exists a facsimile
for Lie Groups [9]:

$$K_{\mathcal{M}}^{\mathrm{SOP}}(x|x_0,t) = \frac{e^{t\rho^2}}{(4\pi t)^{\dim \mathcal{M}/2}} \sum_{2\pi n \in \Gamma} \prod_{\alpha \in R^+} \left( \frac{\alpha \cdot (h+2\pi n)}{2\sin(\alpha \cdot h/2)} \right) e^{-\frac{(h+2\pi n)^2}{4t}} \tag{21}$$

Here, $\Gamma$ is the set of all points in the tangent space to the identity which exponentiate back to the
source point (e.g. in spheres this is all distances which integral multiples of $2\pi$), and $\rho^2$ is a manifold
specific constant. We note that the product over $R^+$ is exactly the $\Delta$ change in variables above since
$m_\alpha = 1$, but we simply extend this to every other root.

Generally, this formula is rather powerful as it gives us an exact (albeit infinite) representation for the
heat kernel. Compared to the eigenfunction expansion in Equation 13, the Sum Over Paths represen-
tation is accurate for small $t$, which nicely complements the fact that the eigenfunction representation
is accurate for large $t$. This formula does generalize to split-rank Riemannian symmetric spaces like
odd dimensional spheres. However, we did not pursue these formulas further since the formulas are
much more complex due to the appearance of intertwining operators.

### 3.2.3  A Unified Heat Kernel Estimator

We unify these approximants into a single heat kernel estimator. Our computation method splits
up the heat kernel evaluation based on time steps, and applies an eigenfunction summation or an
improved small time approximation accordingly. This allows us to effectively control the errors at
both the small and large time steps while significantly reducing the number of function evaluations.
Our full algorithm is outlined in Algorithm 1.

---

**Algorithm 1:** Heat Kernel Computation

---

**Hyperparameters:** Riemannian symmetric space $\mathcal{M}$, number of eigenfunctions $n_e$, time value
  cutoff $\tau$, (optional, depending on if $\mathcal{M}$ is a Lie Group) number of paths $n_p$
**Input:** source $x_0$, time $t$, query value $x$
Compute
**if** $t < \tau$ **then**
  **if** $\mathcal{M}$ is a Lie Group **then**
  │  **return** $K_{\mathcal{M}}^{\mathrm{SOP}}(x|x_0,t)$ truncated to $|n| < n_p$ in the summation over $\Gamma$.
  **else**
  │  **return** $K_{\mathcal{M}}^{\mathrm{SD}}(x|x_0,t)$.
  **end**
**else**
  │  **return** $K_{\mathcal{M}}^{\mathrm{EF}}(x|x_0,t)$ truncated to $|n| < n_e$.
**end**

---

We can compute $\nabla_x \log K_{\mathcal{M}}$ using conventional autodifferentiation tools. As this is the score quantity

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

 and the theoretical framework (see Figure 2). We found that, although RFM is able to easily learn relatively simple distributions, this non-smoothness is highly detrimental for more nontrivial densities and can be crippling in high dimensions (see Figure 3). We also note that, similar to the Euclidean case, RFM with the diffusion path corresponds to score matching with the probability flow ODE, so our work provides a computation for this path.

## 5  Experiments

### 5.1  Simple Test Tasks

We start by comparing our accurate denoising score matching objective with the inaccurate version suggested by [4] based on 50 eigenfunctions. We test on the compiled Earth science datasets from [38], detailing results are in Table 1. Generally, our accurate heat kernel results in a substantial improvement and matches sliced score matching. Note that we do not expect our method to outperform sliced score matching since these datasets are low dimensional.

We also compare directly with RFMs on a series of increasingly complex checkerboard datasets on the flat torus. These datasets have appeared in prior work to measure model quality [3, 5, 37]. As a result of the non-smooth vector field dynamics, we find that RFMs degrade in performance as the checkerboard increases in complexity, and is unable to learn past a certain point. Our visualized results are given in Figure 3.

### 5.2  Learning the Wilson Action on $SU(3)$ Lattices.

We apply our method to learn $SU(3)$ configurations on a $4 \times 4$ lattice. In particular, we generate data on $SU(3)^{4 \times 4}$ according to the Wilson Action [53] $p(K) \propto e^{-S(K)}$, where $S(K)$ is defined as:

$$S(K) = -\frac{\beta}{3} \sum_{x,y \in \mathbb{Z}_4^2} \operatorname{Re} \operatorname{tr}(K_{x,y} K_{x+1,y} K^*_{x+1,y+1} K^*_{x,y+1}) \tag{23}$$

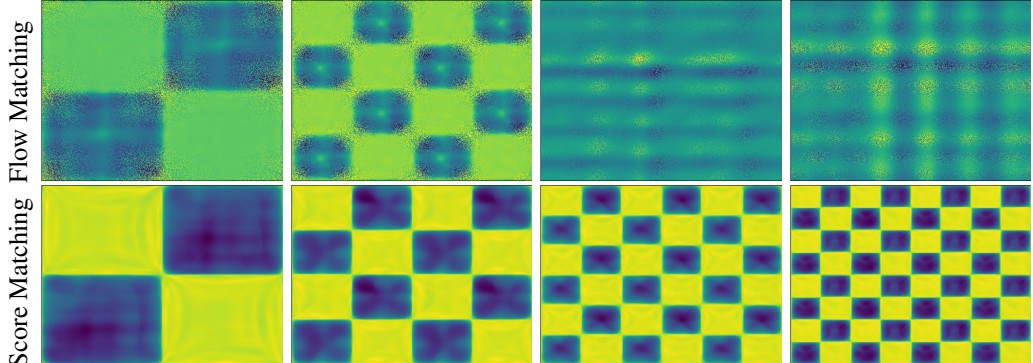

Figure 3: **We compare Riemannian score matching and flow matching on increasingly complex checkerboard patterns on the torus.** On simple checkerboards, Flow matching learns suboptimal distributions with noticeable artifacts like blurriness and spurious peaks, and fails for more complex checkerboards. Conversely, Riemannian score matching/diffusion learns accurate densities.

| Method | SVHN | Places365 | LSUN | iSUN | Texture |
|---|---|---|---|---|---|
| SSD+[45] | 31.19 | 77.74 | 79.39 | 80.85 | 66.63 |
| KNN+[50] | 39.23 | 80.74 | 48.99 | 74.99 | 57.15 |
| CIDER (penultimate layer) | 23.09 | 79.63 | 16.16 | 71.68 | 43.87 |
| CIDER (hypersphere) | 93.53 | 89.92 | 93.68 | 89.92 | 92.41 |
| CIDER (hypersphere) + Diffusion | 28.95 | 76.54 | 35.78 | 74.17 | 62.87 |

Table 2: **We compare contrastive learning OOD detection methods on CIFAR-100.** We report false positive rates (↓) for 0.05 false negative rate. The hyperspherical embeddings produce very bad results, but with a Riemannian Diffusion Model, it is competitive with or surpass the state of the art.

We take $\beta = 9$ for our purposes. Our model trains stably and can learn the density, achieving an ESS of 0.62. This is not the standard variational inference training procedure [6], since it requires samples to train with, but