# OpenReview forum: "Scaling Riemannian Diffusion Models"
_NeurIPS.cc/2023/Conference — NeurIPS 2023 poster_

### Official Review · Reviewer_VM6C · 2023-06-24

**Soundness:** 2 fair
**Presentation:** 2 fair
**Contribution:** 2 fair
**Rating:** 5
**Confidence:** 3

**Summary:**

This paper considers the diffusion model on Riemann space. They propose several strategies for numerically computing the heat kernel on Riemannian symmetric spaces. They also propose two techniques to sample from their proposed heat kernel approximations, including a rejection sampling method and a probability flow method.

**Strengths:**

They compare several techniques to approximate the heat kernel in Riemannian symmetric space.

They propose two techniques to obtain samples from the approximated heat kernel.

They scale the models to the task of 128 dimension, which seems to be promising.

**Weaknesses:**

The proposed several techniques in section 3 seem quite standard, and do not involve much novelty.

From Figure 1, these several heat kernel estimators could be very sensitive, so the hyper-parameter $\tau$ in Algorithm 1 needs very careful tuning.

The experimental parts consider only few baselines. For example, in figure 3 and table 1, can you compare with more previous Riemannian diffusion model works?

**Questions:**

Rejection sampling is known to be non-scalable (low acceptance rate.) in high dimension. Do you have this problem when you apply Cheap Rejection Sampling Methods in section 4?

Can you compare the two techniques in section 4? Which one is more convenient to use in practice?

**Limitations:**

.

---

> ### Author Rebuttal · Authors · 2023-08-09
>
> We thank the reviewer for their review and hope to address any concerns the reviewer may have.
>
> * On novelty of section 3 techniques: we do not claim novelty for the heat kernel computation methods, as these have existed in the physics and applied mathematics fields for decades. Rather, as mentioned by reviewer 1Bu5, our primary contribution is a method to adapt these methods for use in the machine learning community. In particular, our method of combining the approximations is uniquely required for diffusion model training, as it needs fast, accurate, and repeated computation while physicists only need to work with approximate ansatze like the Schwinger-Dewitt approximation. In doing so, we aim to alleviate pre-existing issues in Riemannian diffusion model training, which is an important problem.
> * On tuning $\tau$: we agree that tuning $\tau$ is a sensitive task. However, we emphasize that this needs to be done **once** per manifold. Additionally, we will release our heat kernel computation code (and thus our tuned heat kernel values), which will make adoption straightforward.
> * On baselines: for table 1, we compare with [3] only as we want to study the effects of our heat kernel computation on training. For figure 3, we are comparing directly with [22] to show that continuity in the flow path matters. We are not aware of any other Riemannian diffusion works which we can compare against, as the underlying machinery should effectively be the same as [3].
> * On rejection sampling: for our tasks (and these types of tasks in general), the maximal torus dimension is rather low, which makes rejection sampling quite tractable. So, we did not have problems applying it to many of our use cases, although $S^{127}$ has some other issues despite having a one dimensional maximal torus.
> * On comparing techniques: rejection sampling is more convenient and quick to use in practice, while ODE sampling is more cumbersome and slow but works on places where rejection sampling fails (such as $S^{127}$ and in theory manifolds with higher dimensional maximal tori).

---

> > ### Comment · Reviewer_VM6C · 2023-08-14
> > **Reply**
> >
> > Thank you for your reply! I have a follow-up question: Have you considered comparing with non-diffusion based methods, such as the one blow?
> >
> > Cohen S, Amos B, Lipman Y. Riemannian convex potential maps. In International Conference on Machine Learning 2021 Jul 1 (pp. 2028-2038). PMLR.

---

> > > ### Author Response · Authors · 2023-08-14
> > > **Reply**
> > >
> > > Good question!
> > >
> > > For table 1 and figure 3, our primary purpose was to compare against established diffusion-like methods (in particular simulation-free differential equation methods). In particular, our goal is to not compare against other Riemannian generative modeling techniques, which have already been done in Riemannian diffusion papers [3].

---

> > > > ### Comment · Reviewer_VM6C · 2023-08-17
> > > >
> > > > Thank you! I'll update my score.

---

### Official Review · Reviewer_611J · 2023-06-25

**Soundness:** 4 excellent
**Presentation:** 3 good
**Contribution:** 4 excellent
**Rating:** 7
**Confidence:** 3

**Summary:**

Developing diffusion model on Riemannian manifolds has received attention recently. However, the current proposals did not scale well to high dimension as they used sliced-score matching or denoising score matching with bad approximations of the heat kernel. This work focuses on using better approximations of the heat kernel on (compact) symmetric Riemannian manifolds. Then, they show the benefit of their proposals on several manifolds such as the torus, $S^2$, $S^{127}$ or $SU(n)$, and apply it to different problems such as density estimation of earth data or on the torus, or for contrastively learned hypersperical embeddings.

**Strengths:**

Developing efficient generative models on Riemannian manifolds is of much interest, and this paper proposes to use efficient approximations of the transition kernel, which is pretty appealing.

- Different kernel approximations with an analysis on the setting in which they work or not
- Very competitive results compared to previous methods, especially on high-dimensional manifolds
- Good Applications

**Weaknesses:**

- I find the introduction of the transition kernel a bit quick and it is thus not very clear how to relate it to the equation (7,8) in the Euclidean diffusion. Maybe it would be more clear rewrite the equations with the kernel.
- The experiment in Section 5.2 is not very clear.

**Questions:**

The results of the experiment in Section 5.2 are not clear to me as it only provides an ESS of 0.62, but without any comparison or value of any other method. Thus, I do not understand what it brings.

Not very related to the point of the paper, but it is stated that for non compact Riemannian manifolds, one can "map points to surfaces and learn the diffusion on $\mathbb{R}^d$. Is it not a bit naive? I would expect such method to be less performant than by directly applying the diffusion on the manifold.


Typos:
- Equation 10: lack closing parenthesis
- Line 109: "prior to work to"
- The notation "cot" in equation (15) is not defined.
- Line 288: the abreviation ESS is not defined

**Limitations:**

Yes

---

> ### Author Rebuttal · Authors · 2023-08-09
>
> We would like to thank the reviewer for their positive comments!
>
> * On transition kernel: we will update our final version of the paper to include a more detailed explanation for how the transition kernel is derived from the SDE. In particular, we will emphasize that it comes from considering the probability distribution starting from $x_0$ and following the SDE up to time $t$.
> * On experiment 5.2: the purpose of experiment 5.2 is to show that our model can accurately model complex data on more exotic manifolds $SU(n)$ in high dimensions. Our ESS score is competitive with previous results [5] but we are forced to use an incomparable training algorithm due to the scope of the paper. As mentioned in our response to reviewer NaGF, we believe that combining our results with other advances in diffusion modeling and a more principled architecture design is a promising avenue of further research.
> * On mapping from Euclidean space to noncompact Riemannian symmetric spaces: previous work has shown that learning a generative model on the Euclidean space and mapping back produces very good results and can even outperform learning on the manifold directly since both spaces are diffeomorphic [32]. Generally, we believe that the power of the generative model should be enough to overcome most metric distortion issues that arise from such a mapping procedure. Furthermore, since simpler spaces give much more power for computation and modeling capabilities, this is a trade we should take for our scaling concerns.

---

### Official Review · Reviewer_1Bu5 · 2023-07-02

**Soundness:** 4 excellent
**Presentation:** 4 excellent
**Contribution:** 3 good
**Rating:** 6
**Confidence:** 4

**Summary:**

The theory of stochastic differential equations on vector spaces forms the basis for recent diffusion generative models, and recent efforts have attempted to extend this framework to stochastic differential equations on Riemannian manifolds (Riemannian diffusion models). From a practical computational perspective this extension is less than straightforward as pointed out in this paper, e.g., computing things like "straight lines" (minimal geodesics) become very difficult two-point boundary value problems; quantities that admitted closed-form analytic formulas in the vector space setting do not translate to closed-form formulas in the Riemannian setting. Past attempts at practical work-arounds and computational approximations are limited to low-dimensional cases; these methods do not scale well to higher-dimensional spaces. By restricting the focus to Riemannian manifolds that are also symmetric spaces, the authors present a collection of computation and approximation techniques that, taken together, demonstrate both computational tractability and scalability.  Experiments validating these claims are presented.

**Strengths:**

The paper is technically sound and well-written, and the contributions are laid out in a clear and concise manner without claiming too much. Rather than making claims about the significance of a particular result, I appreciate that the authors claim that it is the sum of a set of smaller contributions that, taken together, lead to a computationally practical, scalable set of techniques to make Riemannian diffusion models workable in practice. The focus on Riemannian symmetric spaces seems reasonable, and the detailed computation formulas and approximations are laid out in enough detail that they can be implemented by a reader familiar with stochastic differential equations and some Riemannian geometry. The examples and experiments appear to support the claims of the authors about computational tractability and scalability to higher-dimensional problems.


**Weaknesses:**

-The following is a general comment applicable not only to this paper, but to all papers that present a “geometric” version of an existing algorithm or method in machine learning. In the past, developing a geometric, manifold version of a vector space algorithm could be regarded as a meaningful contribution in itself, and I would argue that the Riemannian diffusion model is somewhat in this spirit (equivariant models I would argue are not in this spirit however). Of course, it’s hard to argue against the claim that the Riemannian diffusion model is needed for problems in which the underlying data are manifold-valued; in that case I would be much more convinced by examples and case studies drawn from mainstream applications rather than narrow ones.  As geometric methods have become more mainstream, the threshold for what constitutes a meaningful contribution is justifiably higher: given the much more difficult computations involved in computing, e.g., derivatives, gradients, Laplacians, minimal geodesics, kernels, etc., the considerable increase in computation needs to be justified by results. For this paper, the authors recognize and point out the difficulty of computing the geometric quantities and propose more efficient approximations, which is worthwhile (but whether they deserve to be published in NeurIPS is another matter). It would be helpful if the authors can address this question more directly – are the added computational difficulties justified by a commensurate improvement in the results? The experiments do not strike me as mainstream problems, for one thing. (Note: This comment could also have been placed in the "questions" section, but I place it here as it could be a potential weakness).
-The definition of symmetric space could be sharpened, as this is an important underlying assumption in the paper, e.g., clarify the isometry requirement (isometry between what spaces?), make the distinction between local vs global symmetric space, provide brief intuition (reflexive symmetry about a point), and most importantly, provide examples of spaces arising in ML applications that are symmetric (the authors list some -- a few more would be helpful, e.g., the space of symmetric positive-definite matrices, which is nonimpact but a space that arises constantly in ML) and also examples of non-symmetric spaces (e.g., hyperbolic manifolds).
-The assumption of the Riemannian manifold being embedded in a higher-dimensional ambient Euclidean space in itself is not restrictive thanks to Whitney and Nash, and most practical problems that I've encountered usually admit some natural embedding. The compactness assumption, however, could be somewhat restrictive for certain problems in which the underlying data manifold is potentially unbounded. The authors brush aside this case by asserting that noncompact Riemannian manifolds are diffeomorphic to Euclidean space, and therefore the standard way of using the embedding in R^n to model diffusion, then mapping back to the surface, is sufficient. One could then ask why this approach doesn't work for any manifold embedded in Euclidean space; it would be helpful if the authors could clarify this point.

**Questions:**

-Explaining in more detail the Riemannian exponential map would be helpful, so that the reader has a better idea of what the computation entails.
-The examples are for SU(3) and the hypersphere, which are rather specific manifolds. Particularly in the case of the hypersphere, I suspect that the heavy Riemannian diffusion machinery may not be needed to arrive at the result, since Brownian motion on the sphere is well-characterized and projections to the hypersphere are trivial.

**Limitations:**

No potential negative societal impact beyond that of other typical NeurIPS submissions that I can detect.

---

> ### Author Rebuttal · Authors · 2023-08-09
>
> We would like to thank the reviewer for their thorough and interesting comments. We are glad that the reviewer appreciated our presentation, and we hope to address any concerns.
>
> ## Addressing weaknesses:
> * On results: we emphatically agree that showing real-world applications of geometric machine learning methods is highly important, and this was a large motivation behind our work (since previous Riemannian diffusion methods do not currently scale to these real world tasks). We also agree that, while our $SU(n)$ example could be useful for lattice QCD systems down the line, it is a bit niche for the ML community. However, we argue that our contrastive embedding experiment is highly relevant to the broader ML community and relies on Riemannian geometry. These embeddings, which include systems like CLIP, are very mainstream and use the hypersphere as an embedding space (and thus is a naturally Riemannian problem) since it produces significantly better results. Much prior work has shown that these embeddings are not usable for many tasks (see [1] Section 3.2), which we argue is due to the weaker machine learning toolkit available on the sphere. Our experiment tests this phenomena on embedding-based OOD detection (which is a common application area) and shows that we can extract a meaningful representation to get nontrivial numerical results.
> * On definition of symmetric space: we agree that our definition of symmetric spaces (as well as some of the technical constructions that we use for our method) was a bit lackluster, and we will definitely update our presentation for the final version of our paper. However, we do note that hyperbolic space is symmetric.
> * On Euclidean mapping of compact spaces: For non-diffeomorphic manifolds any such mapping results in an extremely high distortion even if we could disregard the discontinuity. Previous work [3, 33] has shown that this tends to greatly hamper results. By contrast, [32] has shown that mapping noncompact Riemannian symmetric spaces to Euclidean space produces good results since the distortion is manageable. For diffusion models, the added computational tractability greatly outweighs this minor distortion which can be overcome with the power of the neural network.
>
> ## Answering questions:
> * On the Riemannian exponential map: certainly! We will update our paper to include formulas for the Riemannian exponential map as well as a discussion of computational details.

---

> > ### Comment · Reviewer_1Bu5 · 2023-08-15
> >
> > Just a short note to confirm that I've read your responses to my queries, thank you. Re your point about CLIP and the hypersphere, I suppose I was looking for other more generic examples other than the hypersphere (all of the Riemannian geometric machinery developed for diffusions on general manifolds typically reduces to very simple notions on the hypersphere, and often can be derived by appealing to more intuitive concepts).

---

### Official Review · Reviewer_NaGf · 2023-07-03

**Soundness:** 3 good
**Presentation:** 3 good
**Contribution:** 3 good
**Rating:** 6
**Confidence:** 4

**Summary:**

In this work the authors leverage properties of symmetric spaces to get better approximation of the heat kernel and therefore of the denoising score matching loss for training diffusion models on compact manifolds.
In particular, using the concept of maximal torus and roots of the symmetric space, the Laplacian can be written in terms of the radial Laplacian which itself depends on these roots and their multiplicity.
Most interestingly, for (compact) Lie groups, the heat kernel can also be written as a summation over all possible paths between the origin and evaluation points, similarly to the heat kernel expression on the torus.
Additionally, for small time scale, the authors propose to use the Schwinger-Dewitt approximation of the heat kernel which can be seen (I think) as a higher order Taylor approximation than the Varadhan one, and therefore is accurate for much higher values of $t$ while staying computationally fast.
The authors then show that these approximations are crucial for training diffusion models on 'high' dimensional manifolds, but still improves performance on smaller ones such as $\mathcal{S}^2$ and $\mathrm{SU}(3)$.

**Strengths:**

- I enjoyed reading this paper, as it is well organised, sounds and generally well written.

- I was not aware of the result from Equation 21 'Sum Over Paths', and have been wondering in the past why does the approximation from a truncation of Eq 13 was getting worse as $t$ gets smaller, while it is the opposite for the heat kernel of the torus. This is really interesting, and would like to know more about the generalisation mentioned at line 219.
Figure 1 is quite illustrative in what the Schwinger-Dewitt approximation brings compared the Varadhan!

- The exact sampling from the (marginal) Brownian motion is also neat.


**Weaknesses:**

- Section 3.1: Would be useful to give some intuition to the Definition 3.1 of symmetric manifolds. Generally, the concept of maximal torus and Equation 15 deserve a better presentation. The 'positive roots of our symmetric space' are not defined. Would be useful to say how the 'radial' Laplacian relates to the Laplacian of this symmetric space, and similarly for its heat kernel to the heat kernel of the symmetric space.
- Table 1: Would be worth including [22] as they overperform [3] on several datasets.
- Flow matching: Why not including the results from Flow Matching [9]? They seem to report better results on the Volcano and Fire datasets. Why does flow matching fails on the chequerboard on the torus but not on the sphere? Isn't the vector field also discontinuous there?


**Questions:**

- Equation 11: As stated in the paragraph below this noising process is indeed for compact manifolds, yet [3] more generally proposed an 'Ornstein-Uhlenbeck' like process for non-compact manifolds, which simplifies to Eq 11 when compact. Would be worth having this more general setting in Section 2.2?
- line 105: Perhaps slightly misleading to say that the heat kernel has 'has no closed form' since it is given by the Sturm–Liouville decomposition of Equation 13, although it is a converging series, and has such not 'computable' (since there is generally no finite time algorithm).
- Line 181: It is indeed good to stress that one can bypass directly computing the eigenfunctions, although I don't think that in the past people have done this (naive approach)? E.g. equation 18 can be found in appendix F of [3].
- Section 3.2.2: Worth linking this with the $\exp(-\lambda_i t)$ term in Eq. 13, so for small values of $t$, terms in the series would tend to contribute more and more equally.
- Eq 20: What is $v$? Where does this come from?
- Figure 1: Would you know why the Schwinger-Dewitt approximation has this very fast change in accuracy?
- 'RFM introduces several geodesic-based vector fields, but these break the smoothness assumption and the theoretical framework'. Could you please expand on that? This is using which (conditional) vector field?
- Section 3.3: This is a really neat idea. Can the ratio be trivially bounded? Worth adding this in the paper! Does this help during training (for the Monte Carlo estimator of the denoising score matching loss)?

**Limitations:**

- This paper 'do not consider the case of noncompact Riemannian symmetric spaces, as these are diffeomorphic to Euclidean space'. I agree that since the topology is Euclidean it is perhaps less crucial to parameterise a probabilistic model directly on that space, yet the geometry is still different and only locally linear, so the geometry is more and more distorted further away, e.g. at boundary for the Poincare model of hyperbolic geometry. Is there some equivalent notion of the maximal Torus for non compact manifolds?
- Would be worth expanding the evaluations in Section 5.2 which is a really interesting setting.

---

> ### Author Rebuttal · Authors · 2023-08-09
>
> We would like to thank the reviewer for their detailed review and insightful comments. We are glad that the reviewer found the paper enjoyable to read, and we hope to address any concerns the reviewer has.
>
> * On the generalization to spheres: we will include the generalization to split-rank symmetric spaces e.g. odd dimensional spheres. Effectively, the summation in Equation (18) must be altered to include an additional function of $n$ (see [8]). The computation of this function takes $O(n)$ extra compute but can be vectorized, and it is best to think of it as an extension to the Schwinger-Dewitt approximation.
>
> ## Addressing weaknesses
>
> * On section 3.1: we agree with the reviewer that our presentation of these topics was a bit rushed. We will update our final paper to better present these concepts.
> * On [22] compared to [3] in Table 1: [22] uses a very similar training method to [3], but adds in extra components such as variance minimization. For our comparison, we compare only against [3] because it is barebones (and thus has the fewest variables).
> * On Flow Matching: we don’t include RFM results in Table 1 as we are not aiming for SOTA on these datasets; rather, we want to conduct a comparison with minimal confounding variables. RFM also has the discontinuity on the sphere, but this doesn’t seem to matter since the data points have fewer modes and a smaller support. However, we have found that RFM tends to quickly degrade when the task becomes harder (through dimension or data), which hampers applications in real world examples.
>
> ## To answer questions
>
> * Equation 11: From what we can see, both [3] and [22] both address the noncompact case using an OU process of the form $dx = -\log_0(x) dt + dB_t$ (resulting in a limiting distribution of wrapped Gaussian). The issue with these processes is that their transition kernels are not known (and aren’t even heat kernels like the standard OU process). Alternatively, we could apply a noncompact heat kernel computation method, but this would require using a VESDE which is known to be suboptimal for likelihoods since the distribution does not mix.
> * Line 105: We disagree and believe that closed form is an accurate description, since the infinite summation violates the requirements for “closed form”.
> * Line 181: We definitely agree! We wanted to emphasize in the paper that, for the Riemannian symmetric spaces that we consider, this trick of collapsing eigenfunction summations is broadly generalizable, and we will update our paper to better reflect this.
> * Section 3.2.2.: We agree and will update our paper.
> * Eq 20: Apologies, this was supposed to be $h$.
> * Figure 1: We think that Schwinger-Dewitt approximation error is exponential (which accounts for the large jump), but smooths out as the heat kernel stabilizes for large time steps.
> * RFM vector field: certainly! The geodesic path conditional vector field is given by $\exp_x(t \log_x(y))$ where $x$ and $y$ are the noise and the data point respectively. The issue is that $\log_x(y)$ is not defined when $y$ is the cut locus of $x$, and in particular it is highly nonsmooth near these points as well. For example, for spheres we can imagine that when $y$ is opposite $x$ then the vector field $\log_x(\cdot)$ is a bunch of large diverging tangent vectors when evaluated around $y$.
> * Section 3.3: The ratio can be trivially bounded for most manifolds, and we found that the main benefit for our simulation strategies was a noticeable speed increase when compared with the many SDE Riemannian exponential map simulation steps that previous methods have to use to get good results [3, 22], which matters when working with many dimensions.
>
> ## Addressing Limitations
>
> * On mapping to Euclidean space for noncompact manifolds: we believe that the additional computational benefits we get from working on a simpler space should outweigh potential metric issues, especially since our neural network models are particularly powerful.
> * On analogues for noncompact manifolds: we will update our manuscript with the analogous heat kernel derivations for noncompact manifolds. The maximal torus is replaced with the maximal abelian subgroup, and we can derive a “sum over paths” representation for the heat kernel when the manifold is dual to a valid compact case. In particular, this is a singular sum (as there is only one path between any two points). This works for the SPD manifold (as it is dual to SO(n)) and odd-dimensional hyperbolic space (which is dual to odd-dimensional spheres).
> * On section 5.2: we agree, but this may be a bit out of scope for our current paper. Doing so requires a more principled architectural design as well as a significantly different training procedure [Cite1]. Our current results are there to mainly show that our method can learn the complex data on the more exotic $SU(n)$ manifold at these scales.
>
> [Cite1] https://arxiv.org/pdf/2211.01364.pdf

---

> > ### Comment · Reviewer_NaGf · 2023-08-18
> > **response**
> >
> > Thanks for your detailed response!
> >
> > > we have found that RFM tends to quickly degrade when the task becomes harder (through dimension or data), which hampers applications in real world examples.
> >
> > Re the checkerboard from Figure 3: I am still struggling to understand how come the (optimal) flow suffers from this 'lack of smoothness' more than the Stein score. Would you please be able to expand on this?
> >
> > > Section 3.3: The ratio can be trivially bounded for most manifolds, and we found that the main benefit for our simulation strategies was a noticeable speed increase when compared with the many SDE Riemannian exponential map simulation steps that previous methods have to use to get good results [3, 22], which matters when working with many dimensions.
> >
> > That's cool!
> >
> > > On section 3.1: we agree with the reviewer that our presentation of these topics was a bit rushed. We will update our final paper to better present these concepts.
> >
> > That is really important indeed.

---

### Decision · Program_Chairs · 2023-09-21

**Decision:**

Accept (poster)

**Comment:**

This paper presents several practical improvements for scaling up Riemannian Diffusion Models on Riemannian symmetric spaces, leading in particular to an improved heat kernel estimator. The proposed method is accompanied by numerical experiments.

Reviewers generally agree that the paper is technically sound, well-written, and of practical interest.

The authors should take the comments from the reviewers into account to improve the presentation of the manuscript.